# An Improved Relaxation for Oracle-Efficient Adversarial Contextual Bandits

**Kiarash Banihashem**
University of Maryland, College Park
kiarash@umd.edu

**MohammadTaghi Hajiaghayi**
University of Maryland, College Park
hajiagha@umd.edu

**Suho Shin**
University of Maryland, College Park
suhoshin@umd.edu

**Max Springer**
University of Maryland, College Park
mss423@umd.edu

## Abstract

We present an oracle-efficient relaxation for the adversarial contextual bandits problem, where the contexts are sequentially drawn i.i.d from a known distribution and the cost sequence is chosen by an online adversary. Our algorithm has a regret bound of $O(T^{\frac{2}{3}}(K \log(|\Pi|))^{\frac{1}{3}})$ and makes at most $O(K)$ calls per round to an offline optimization oracle, where $K$ denotes the number of actions, $T$ denotes the number of rounds and $\Pi$ denotes the set of policies. This is the first result to improve the prior best bound of $O((TK)^{\frac{2}{3}}(\log(|\Pi|))^{\frac{1}{3}})$ as obtained by Syrgkanis et al. at NeurIPS 2016, and the first to match the original bound of Langford and Zhang at NeurIPS 2007 which was obtained for the stochastic case.

## 1   Introduction

One of the most important problems in the study of online learning algorithms is the *contextual bandits problem*. As a framework for studying decision making in the presence of side information, the problem generalizes the classical *multi-armed bandits problem* and has numerous practical applications spanning across clinical research, personalized medical care and online advertising, with substantial emphasis placed on modern recommender systems.

In the classical multi-armed bandits problem, a decision maker is presented with $K$ actions (or arms) which it needs to choose from over a sequence of $T$ rounds. In each round, the decision maker makes its (possibly random) choice and observes the cost of its chosen action. Depending on the setting, this cost is generally assumed to be either *stochastic* or *adversarial*. In the stochastic setting, the cost of each action is sampled i.i.d. from a fixed, but a priori unknown, distribution. In the more general adversarial setting, no such assumption is made and the costs in each round can be controlled by an online adversary. The goal of the learner is to minimize its *regret*, defined as the absolute difference between its total cost and the total cost of the best fixed action in hindsight.

The contextual bandits problem generalizes this by assuming that in each round, the learner first sees some side information, referred to as a *context*, $x \in \mathcal{X}$ and chooses its action based on this information. As in prior work (Rakhlin and Sridharan, 2016; Syrgkanis et al., 2016b), we assume that the $x$ are sampled i.i.d. from a fixed distribution $\mathcal{D}$, and that the learner can generate samples from $\mathcal{D}$ as needed. In addition, the learner has access to a set of *policies*, $\Pi$, where a policy is defined as a mapping from contexts to actions. As before, the goal of the learner is to minimize its regret, which we here define as the absolute difference between its total cost and the total cost of the best policy of $\Pi$ in hindsight.

37th Conference on Neural Information Processing Systems (NeurIPS 2023).

It is well-known that by viewing each policy as an expert, the problem can be reduced to the *bandits with experts* problem where, even in the adversarial setting, the EXP4 (Auer et al., 1995) algorithm achieves the optimal regret bound of $O(\sqrt{TK\log(|\Pi|)})$. Computationally however, the reduction is inefficient as the algorithm's running time would be linear in $|\Pi|$. Since the size of the policy set, $\Pi$, can be very large (potentially exponential), the utility of this algorithm is restricted in practice.

Given the computational challenge, there has been a surge of interest in *oracle-efficient* algorithms. In this setting, the learner is given access to an Empirical Risk Minimization (ERM) optimization oracle which, for any sequence of pairs of contexts and loss vectors, returns the best fixed policy in $\Pi$. This effectively reduces the online problem to an offline learning problem, where many algorithms (e.g., SVM) are known to work well both in theory and practice. This approach was initiated by the seminal work of Langford and Zhang (2007), who obtained a regret rate of $O(T^{2/3}(K\log|\Pi|)^{1/3})$ for *stochastic rewards*, using an $\epsilon$-greedy algorithm. This was later improved to the information-theoretically optimal $O(\sqrt{TK\log|\Pi|})$ (Dudik et al., 2011). Subsequent works have focused on improving the running time of these algorithms (Beygelzimer et al., 2011; Agarwal et al., 2014; Simchi-Levi and Xu, 2022), and extending them to a variety of problem settings such as auction design (Dudík et al., 2020), minimizing dynamic regret (Luo et al., 2018; Chen et al., 2019), bandits with knapsacks (Agrawal et al., 2016), semi-bandits (Krishnamurthy et al., 2016), corralling bandits (Agarwal et al., 2017), smoothed analysis (Haghtalab et al., 2022; Block et al., 2022), and reinforcement learning (Foster et al., 2021).

Despite the extensive body of work, progress for *adversarial rewards* has been slow. Intuitively, approaches for the stochastic setting do not generalize to adversarial rewards because they try to "learn the environment" and in the adversarial setting, there is no environment to be learnt. This issue can be seen in the original multi-armed bandits problem as well. While the regret bounds for the adversarial setting and the stochastic setting are the same, the standard approaches are very different. [1] In the stochastic setting, the most standard approach is the UCB1 algorithm (Auer et al., 2002), which is intuitive and has a relatively simple analysis. In the adversarial setting however, the standard approach is considerably more complex: the problem is first solved in the "full feedback" setting, where the learner observes all of the rewards in each iteration, using the Hedge algorithm. This is then used as a black box to obtain an algorithm for the partial (or bandit) feedback setting by constructing unbiased estimates for the rewards vector. The analysis is also much more involved compared to UCB1; a standard analysis of the black box reduction leads to the suboptimal bound of $T^{3/4}K^{1/2}$, and further obtaining the optimal $\sqrt{TK}$ bound requires a more refined second moment analysis (see Chapter 6 of Slivkins (2019) for a more detailed overview).

As a result, oracle-efficient algorithms for the adversarial setting were first proposed by the independent works of Rakhlin and Sridharan (2016) and Syrgkanis et al. (2016a) in ICML 2016, who obtained a regret bound of $O(T^{3/4}K^{1/2}\log(|\Pi|)^{1/4})$ and left obtaining improvements as an open problem. This was subsequently improved to $O((TK)^{2/3}\log(|\Pi|)^{1/3})$ by Syrgkanis et al. (2016b) at NeurIPS 2016.

## 1.1 Our contribution and techniques

In this work, we design an oracle-efficient algorithm with a regret bound of $O(T^{2/3}(K\log|\Pi|)^{1/3})$. Our result is the first improvement after that of Syrgkanis et al. (2016b), and maintains the best regret upper bound to the extent of our knowledge. This is also the first result to match the bound of Langford and Zhang (2007), the original baseline algorithm for the stochastic version of the problem. We state the informal version of our main result (Theorem 7) in the following.

**Theorem 1** (Informal). *For large enough $T$,[2] there exists an algorithm (Algorithm 1) that achieves expected regret on the order of $O(T^{2/3}(K\log|\Pi|)^{1/3})$ for the adversarial contextual bandits problem using at most $O(K)$ calls per round to an ERM oracle.*

In order to compare this result with prior work, it is useful to consider the regime of $K = T^\alpha$ for a constant $\alpha > 0$. In this regime, our work leads to a sublinear regret bound for any $\alpha < 1$, while

---

[1]We note that any approach for the adversarial setting works for the stochastic setting as well. However, both in theory and practice, specialised approaches are more commonly used.

[2]When $T$ is small, our improvement compared to prior work is more significant; we focus on the large $T$ setting to obtain a fair comparison.

prior work (Rakhlin and Sridharan, 2016; Syrgkanis et al., 2016b) can only obtain sublinear regret for $\alpha < 1/2$.

Our improved dependence on $K$ is important practically since, for many real-world implementations such as recommender systems, the number of possible actions is very large. Additionally, many bandits algorithms consider "fake" actions as part of a reduction. For example, a large number of actions may be considered as part of a discretization scheme for simulating a continuous action space (Slivkins, 2009). In such cases, the improved dependence on $K$ could potentially imply an overall improvement with respect to $T$, as the parameters in the algorithm are often chosen in such a way that optimizes their trade-off.

From a technical standpoint, our result builds on the existing works based on the relax-and-randomize framework of Rakhlin et al. (2012). Rakhlin and Sridharan (2015) used this framework, together with the "random playout" method, to study online prediction problems with evolving constraints under the assumption of a full feedback model. Rakhlin and Sridharan (2016) extended these techniques to the partial (bandit) feedback model, and developed the BISTRO algorithm which achieves a $O(T^{3/4}K^{1/2}\log(|\Pi|)^{1/4})$ regret bound for the adversarial contextual bandits problem. Subsequently, Syrgkanis et al. (2016b) used a novel distribution of hallucinated rewards as well as a sharper second moment analysis to obtain a regret bound of $O((TK)^{2/3}\log(|\Pi|)^{1/3})$. We further improve the regret rate to $O(T^{2/3}(K\log(|\Pi|))^{1/3})$ by reducing the support of the hallucinated rewards vector to a single random entry. We note the previous approaches of Rakhlin and Sridharan (2016); Syrgkanis et al. (2016b), as well as the closely related work of Rakhlin and Sridharan (2015), set all of the entries of the hallucinated cost to i.i.d Rademacher random variables.

We show that our novel relaxation preserves the main properties required for obtaining a regret bound, specifically, it is *admissible*. We then prove that the Rademacher averages term that arises from our new relaxation improves by a factor of $K$, which consequently leads to a better regret bound. We further refer to Section 3 for a more detailed description of our algorithm, and to Section 4 for the careful analysis.

## 1.2    Related work

**Contextual bandits.** There are three prominent problem setups broadly studied in the contextual bandits literature: Lipschitz contextual bandits, linear contextual bandits, and contextual bandits with policy class. Lipschitz contextual bandits (Lu et al., 2009; Cutkosky and Boahen, 2017) and linear contextual bandits (Chu et al., 2011) assume a structured payoff based on a Lipschitz or linear realizability assumption, respectively. The strong structural assumptions made by these works however make them impractical for many settings.

To circumvent this problem, many works consider contextual bandits with policy classes where the problem is made tractable by making assumptions on the benchmark of regret. In these works, the learner is given access to a policy class $\Pi$ and competes against the best policy in $\Pi$. This approach also draws connections to offline machine learning models that in recent years have had a huge impact on many applications. In order for an algorithm to be useful in practice however, it needs to be computationally tractable, thus motivating the main focus of this work.

**Online learning with adversarial rewards.** Closely related to our work is the online learning with experts problem where, in each round, the learner observes a set of $N$ experts making recommendations for which action to take, and decides which action to choose based on these recommendations. The goal of the learner is to minimize its regret with respect to the best expert in hindsight. In the full feedback setting, where the learner observes the cost of all actions, the well-known Hedge (Cesa-Bianchi et al., 1997) algorithm based on a randomized weighted majority selection rule achieves the best possible regret bound of $O(\sqrt{T\ln N})$. Correspondingly, in the partial feedback setting, EXP4 (Auer et al., 1995) exploits Hedge by constructing unbiased "hallucinated" costs based on the inverse propensity score technique, and achieves the optimal regret bound of $O(\sqrt{KT\ln N})$. By considering an expert for each policy, the contextual bandits problem can be reduced to this problem. This reduction suffers from computational intractability however due to the linear dependence on $|\Pi|$ in the running time. Since the number of policies can be very large in practice, this poses a major bottleneck in many cases. We alleviate this intractability issue through a computationally feasible oracle-based algorithm with improved regret bound.

**Oracle efficient online learning.** Stemming from the seminal work of Kalai and Vempala (2005), there has been a long line of work investigating the computational barriers and benefits of online learning in a variety of paradigms. Broadly speaking, the bulk of online algorithms are designed on the basis of two popular frameworks in this literature: follow-the-perturbed-leader (Kalai and Vempala, 2005; Suggala and Netrapalli, 2020; Dudík et al., 2020; Haghtalab et al., 2022) and relax-and-randomize (Rakhlin et al., 2012). Both frameworks aim to inject random noise into the input set before calling an oracle to construct a more robust sequence of actions to be played against an adversary, but differ in how they introduce such noise to the system. Our algorithm builds on the relax-and-randomize technique and improves upon the previous best result of Syrgkanis et al. (2016b).

Despite their computational advantages, it is known that oracle efficient algorithms have fundamental limits and, in some settings, they may not achieve optimal regret rates (Hazan and Koren, 2016). Whether this is the case for the adversarial contextual bandits problem remains an open problem.

## 2 Preliminaries

In this section, we explain the notation and problem setup, and review the notion of relaxation based algorithms in accordance with prior work (Rakhlin and Sridharan, 2016; Syrgkanis et al., 2016b).

### 2.1 Notation and problem setup

Given an integer $K$, we use $[K]$ to denote the set $\{1, \ldots, K\}$ and $a_{1:K}$ to denote $\{a_1, \ldots, a_k\}$. We similarly use $(a, b, c)_{1:K}$ to denote the set of tuples $\{(a_1, b_1, c_1), \ldots, (a_K, b_K, c_K)\}$. The vector of zeros is denoted as $\mathbf{0}$, and similarly, the vector of ones is denoted $\mathbf{1}$.

We consider the contextual bandits problem with $[T]$ rounds. In each round $t \in [T]$, a context $x_t$ is shown to the learner, who chooses an action $\hat{y}_t \in [K]$, and incurs a loss of $c_t(\hat{y}_t)$, where $c_t \in [0, 1]^k$ denotes the cost vector. The choice of the action $\hat{y}_t$ can be randomized and we assume that the learner samples $\hat{y}_t$ from some distribution $q_t$. The cost vector is chosen by an adversary who knows the cost vector $x_t$ and the distribution $q_t$ but, crucially, does not know the value of $\hat{y}_t$.

As in prior work (Rakhlin and Sridharan, 2016; Syrgkanis et al., 2016b), we operate in the *hybrid i.i.d-adversarial* model where $x_t$ is sampled from some fixed distribution $\mathcal{D}$, and the learner has sampling access to the distribution $\mathcal{D}$. We additionally assume that the feedback to the learner is partial, i.e., the learner only observes $c_t(\hat{y}_t)$ and not the full cost vector $c_t$.

The learner's goal is to minimize its total cost compared to a set of policies $\Pi$, where a policy is defined as a mapping from contexts to actions. Formally, the learner aims to minimize its *regret*, which we define as

$$\text{REG} := \sum_{t=1}^{T} \langle q_t, c_t \rangle - \inf_{\pi \in \Pi} \sum_{t=1}^{T} c_t(\pi(x_t)),$$

where $\langle q_t, c_t \rangle$ denotes the dot product of $q_t$ and $c_t$, and $\inf$ denotes the infimum.

We assume that the learner has access to a *value-of-ERM* optimization oracle that takes as input a sequence of contexts and cost vectors $(x, c)_{1:t}$, and outputs the minimum cost obtainable by a policy in $\Pi$, i.e., $\inf_{\pi \in \Pi} \sum_{\tau=1}^{t} c_\tau(\pi(x_\tau))$.

### 2.2 Relaxation Based Algorithms

In each round $t \in [T]$ after selecting an action and observing the adversarial cost, the learner obtains an *information tuple*, which we denote by $I_t(x_t, q_t, \hat{y}_t, c_t(\hat{y}_t), S_t)$. Here, $\hat{y} \sim q_t$ is the action chosen from the learner's distribution, and $S_t$ is the internal randomness of our algorithm, which can also be used in the subsequent rounds.

Given the above definition, the notions of *admissible relaxation* and *admissible strategy* are defined as follows.

**Definition 2.** *A partial information relaxation* $\text{REL}(\cdot)$ *is a mapping from the information sequence* $(I_1, ..., I_t)$ *to a real value for any* $t \in [T]$. *Moreover, a partial-information relaxation is deemed*

*admissible if for any such t, and for all $I_1, ..., I_{t-1}$:*

$$\mathbb{E}_{x_t \sim D} \left[ \inf_{q_t} \sup_{c_t} \mathbb{E}_{\hat{y}_t \sim q_t, S_t} \left[ c_t(\hat{y}_t) + \text{REL}(I_{1:t}) \right] \right] \leq \text{REL}(I_{1:t-1}), \tag{1}$$

*and for all $x_{1:T}, c_{1:T}$ and $q_{1:T}$:*

$$\mathbb{E}_{\hat{y}_{1:T} \sim q_t, S_{1:T}} \left[ \text{REL}(I_{1:T}) \right] \geq -\inf_\pi \sum_{t=1}^{T} c_t(\pi(x_t)). \tag{2}$$

*A randomized strategy $q_{1:T}$ is admissible if it certifies the admissibility conditions (1) and (2).*

Intuitively, relaxation functions allow us to decompose the regret across time steps, and bound each step separately using Equation (1). The following lemma formalizes this idea.

**Lemma 3** (Rakhlin and Sridharan (2016)). *Let* REL *be an admissible relaxation and $q_{1:T}$ be a corresponding admissible strategy. Then, for any $c_{1:T}$, we have the bound*

$$\mathbb{E} \left[ \text{REG} \right] \leq \text{REL}(\emptyset).$$

## 3  Contextual Bandits Algorithm

We here define an admissible strategy in correspondence with the relaxation notion from the prior section, and use it to outline our contextual bandits algorithm. As mentioned in Section 1.1, our algorithm is based on the BISTRO+ algorithm of Syrgkanis et al. (2016b), and our improvement is obtained by defining a new relaxation function, which we discuss below. We discuss how this improves the regret bound in Section 4.

**Unbiased cost vectors.** In order to handle the partial feedback nature of the problem, we use the standard technique of forming an unbiased cost vector from the observed entry, together with the discretization scheme of Syrgkanis et al. (2016b). Let $\gamma < 1$ be a parameter to be specified later. Using the information $I_t$ collected on round $t$, we set our estimator to be the random vector whose elements are defined by a Bernoulli random variable

$$\hat{c}_t(i) = \begin{cases} K\gamma^{-1} \cdot \mathbb{1} \left[ i = \hat{y}_t \right] & \text{with probability } \gamma \cdot \frac{c_t(\hat{y}_t)}{K q_t(\hat{y}_t)} \\ 0 & \text{otherwise} \end{cases}. \tag{3}$$

We note that this is only defined for $\min_i q_t(i) \geq \gamma/K$, thus imposing a constraint that must be ensured by our algorithm. It is easy to verify that this vector is indeed an unbiased estimator:

$$\mathbb{E}_{\hat{y}_t \sim q_t} \left[ \hat{c}_t(i) \right] = q_t(i) \cdot \gamma \frac{c_t(i)}{K q_t(i)} \cdot K\gamma^{-1} = c_t(i).$$

**Relaxation function.** We first construct a one-hot Rademacher random vector by randomly sampling an action $i \in [K]$ and setting $\varepsilon_t(j) = 0$ for $i \neq j$ and $\varepsilon_t(i)$ to a Rademacher random variable in $\{-1, 1\}$. We additionally define $Z_t \in \{0, K\gamma^{-1}\}$ that takes value $K\gamma^{-1}$ with probability $\gamma$ and 0 otherwise. Using the notation $\rho_t$ for the random variable tuple $(x, \varepsilon, Z)_{t+1:T}$, we define our relaxation REL as

$$\text{REL}(I_{1:t}) = \mathbb{E}_{\rho_t} \left[ R((x, \hat{c}_t)_{1:t}, \rho_t) \right], \tag{4}$$

where $R((x, \hat{c}_t)_{1:t}, \rho_t)$ is defined to be

$$\gamma(T - t) - \inf_\pi \left( \sum_{\tau=1}^{t} \hat{c}(\pi(x_\tau)) + \sum_{\tau=t+1}^{T} 2Z_\tau \varepsilon_\tau(\pi(x_\tau)) \right).$$

We note the contrast between the above definition and the relaxation frameworks used in prior work (Rakhlin and Sridharan, 2015, 2016; Syrgkanis et al., 2016b): These works all set every entry in $\varepsilon_t$ to a Rademacher random variables, while we set only a single (randomly chosen) entry to a Rademacher random variable and set the rest of the entries to zero.

The changes in the relaxtion function are motivated by the algorithm analysis (see Section 4). Specifically, in order to ensure admissibility, the symmetrization step of the Relax and randomize framework applies only to a single (random) action. Applying noise to all the entries, as is done in

prior work, leads to valid upper bound but is not tight. As we show in Lemma 9, applying noise to a single entry is sufficient, as long as this entry is chosen uniformly at random. The reduced noise leads to an improved Rademacher averages term (see Theorem 6), which in turn leads to a better regret bound.

**Randomized strategy.** As in prior work (Rakhlin and Sridharan, 2015, 2016; Syrgkanis et al., 2016b), we use the "random playout" technique to define our strategy. We use hallucinated future cost vectors, together with unbiased estimates of the past cost, to choose a strategy that minimizes the total cost across $T$ rounds.

Define $D := \{K\gamma^{-1} \cdot \mathbf{e_i} : i \in [K]\} \cup \{\mathbf{0}\}$, where $\mathbf{e_i}$ is the $i$-th standard basis vector in $K$ dimensions. We further define $\Delta_D$, the set of distributions over $D$, and $\Delta'_D \subseteq \Delta_D$ to be the set

$$\{p \in \Delta_D : \max_{i \in [K]} p(i) \leq \frac{\gamma}{K}\}. \tag{5}$$

Recall that $\rho_t$ denotes the random variable tuple $(x, \varepsilon, Z)_{t+1:T}$. We sample $\rho_t$ and define $q_t^*(\rho_t)$ as:

$$q_t^*(\rho_t) := \min_{q \in \Delta_K} \sup_{p_t \in \Delta'_D} \mathbb{E}_{\hat{c}_t \sim p_t} \left[ \langle q, \hat{c}_t \rangle + R((x, \hat{c})_{1:t}, \rho_t) \right]. \tag{6}$$

We than sample the action $\hat{y}_t$ from the distribution $q_t(\rho_t)$ defined as

$$q_t(\rho_t) := (1 - \gamma)q_t^*(\rho_t) + \frac{\gamma}{K} \cdot \mathbf{1}. \tag{7}$$

In order to calculate $q_t(\rho_t)$, we use a water-filling argument similar to Rakhlin and Sridharan (2016) and Syrgkanis et al. (2016b). Formally, we will use the following lemma, the proof of which is in Appendix B.

**Lemma 4.** *There exists a water-filling algorithm that computes the value $q_t^*(\rho_t)$ for any given $\rho_t$ in time $O(K)$ with only $K + 1$ accesses to a value-of-ERM oracle in every round.*

A full pseudocode of our approach is provided in Algorithm 1.

---

**Algorithm 1:** Contextual Bandits Algorithm

---
**for** $t = 1, 2, \ldots, T$ **do**

    Observe context $x_t$

    Draw random variable tuple $\rho_t = (x, \varepsilon, Z)_{t+1:T}$

    Compute $q_t(\rho_t)$ via Equation 7

    Draw action $\hat{y}_t \sim q_t(\rho_t)$ and observe $c_t(\hat{y}_t)$

    Estimate cost vector $\hat{c}_t$ via Equation 3

**end**

---

## 4  Analysis

In this section, we provide the formal statement of our theoretical guarantees and discuss their proofs. Due to space constraints, some of the proofs are deferred to the supplementary material.

As mentioned in the introduction, our main novelty is the use of a new relaxation function, which we discussed in Section 3, that uses less variance in the hallucinated cost vectors. Our initial result verifies that the our novel relaxation is indeed admissible and, as a result, we can leverage the prior work demonstrating the expected regret of these algorithms.

**Theorem 5.** *The relaxation function defined in* (4)*, and the corresponding strategy* (7) *are admissible (Definition 2).*

Theorem 5 contrasts with existing admissible relaxations in that it only uses a single Rademacher variable for each time step, while prior work – Lemma 2 in Rakhlin and Sridharan (2015), Theorem 2 in Rakhlin and Sridharan (2016) and Theorem 3 in Syrgkanis et al. (2016b) – all use $k$ independent Rademacher variables. To our knowledge, this is the first work in which the number of Rademacher variables used in the relaxation does not grow with the number of arms. As we discuss below, this

allows us to reduce the variance of the hallucinated costs, leading to a better regret bound. The proof of Theorem 5 is provided in Section 4.1, and is based on a novel symmetrization step (Lemma 9), which may be of independent interest.

As highlighted in Section 2.2, admissible relaxations are a powerful framework for upper bounding the expected regret in online learning through Lemma 3 and the value of $\text{REL}(\emptyset)$. Formally, Lemma 3 implies

$$\mathbb{E}\left[\text{REG}\right] \leq \text{REL}(\emptyset) = \gamma T + \mathbb{E}_{\rho_0}\left[\sup_{\pi \in \Pi}\left(\sum_{\tau=1}^{T} 2Z_\tau \varepsilon_\tau(\pi(x_\tau))\right)\right]. \tag{8}$$

In order to bound the regret of our algorithm, it suffices to bound the Rademacher averages term above, which we formally do in the following Theorem.

**Theorem 6.** *For any $\gamma > \frac{K}{T}\log(|\Pi|)/2$, the following holds:*

$$\mathbb{E}_{(Z,\varepsilon)_{1:T}}\left[\sup_{\pi \in \Pi}\sum_{i=1}^{T} Z_t \varepsilon_t(\pi(x_t))\right] \leq 2\sqrt{\frac{KT\log|\Pi|}{\gamma}}.$$

The above theorem can be thought of as an improved version of Lemma 2 from Syrgkanis et al. (2016b), where we improve by a factor of $K$. Our improvement comes from the use of the new Rademacher vectors that only contain a single non-zero coordinate, together with a more refined analysis. We refer to Section 4.2 for a formal proof of the result.

Combining Lemma 4, Equation (8), and Theorem 6, we obtain the main result of our paper which we state here.

**Theorem 7.** *The contextual bandits algorithm implemented in Algorithm 1 has expected regret upper bounded by*

$$4\sqrt{\frac{TK\log(|\Pi|)}{\gamma}} + \gamma T,$$

*for any $\frac{K\log|\Pi|}{2T} < \gamma \leq 1$, which implies the regret order of $O((K\log|\Pi|)^{1/3}T^{2/3})$ when $T > 4K\log(|\Pi|)$. Furthermore, the Algorithm makes at most $K+1$ calls to a value-of-ERM oracle in each round.*

We refer to Appendix A for the proof of this result.

## 4.1 Proof of Theorem 5

In order to prove Theorem 5, we need to verify the final step condition (2), and show that the $q_t$ defined in Equation (7) certifies the condition (1), i.e.,

$$\mathbb{E}_{x_t}\left[\sup_{c_t}\mathbb{E}_{\hat{y}_t, S_t}\left[c_t(\hat{y}_t) + \text{REL}(I_{1:t})\right]\right] \leq \text{REL}(I_{1:t-1}), \tag{9}$$

where $\hat{y}_t$ is sampled from $q_t$ and $I_{1:t}$ denotes $(I_{1:t-1}, I_t(x_t, q_t, \hat{y}_t, c_t, S_t))$. Verifying condition (2) is standard and we do this in Appendix D. It remains to prove Equation (9). Since most admissibility proofs in the literature (Rakhlin et al., 2012; Rakhlin and Sridharan, 2015, 2016; Syrgkanis et al., 2016b) follow the framework of the original derivation of Rakhlin et al. (2012), in order to emphasize our novelty, we divide the proof into two parts. The first part (Lemma 8) is based on existing techniques (in particular, the proof of Theorem 3 in Syrgkanis et al. (2016b)) and its proof is provided in Appendix C. The second part (Lemma 9) uses new techniques and we present its proof here.

**Lemma 8.** *For any $t \in [T]$, define $A_{\pi,t}$ and $C_t$ as*

$$A_{\pi,t} := -\sum_{\tau=1}^{t-1}\hat{c}_\tau(\pi(x_\tau)) - \sum_{\tau=t+1}^{T} 2Z_\tau \varepsilon_\tau(\pi(x_\tau)), \quad C_t := \gamma(T-t+1).$$

*Letting $\delta$ denote a Rademacher random variable independent of $\rho_t$ and $\hat{c}_t$, the following holds for any value of $x_t$:*

$$\sup_{c_t}\mathbb{E}_{\hat{y}_t, S_t}\left[c_t(\hat{y}_t) + \text{REL}(I_{1:t})\right] \leq \mathbb{E}_{\rho_t}\left[\sup_{p_t \in \Delta_D'}\mathbb{E}_{\hat{c}_t \sim p_t, \delta}\left[\sup_{\pi \in \Pi}\left(2\delta\hat{c}_t(\pi(x_t)) + A_{\pi,t}\right)\right]\right] + C_t.$$

**Lemma 9.** *Defining $A_{\pi,t}$ as in Lemma 8, the following bound holds for any $t \in [T]$:*

$$\sup_{p_t \in \Delta'_D} \mathbb{E}_{\hat{c}_t \sim p_t, \delta} \left[ \sup_{\pi \in \Pi} \left( 2\delta \hat{c}_t(\pi(x_t)) + A_{\pi,t} \right) \right] \leq \mathbb{E}_{\varepsilon_t, Z_t} \left[ \sup_{\pi \in \Pi} \left( 2Z_t \cdot \varepsilon_t(\pi(x_t)) + A_{\pi,t} \right) \right]. \quad (10)$$

Combining the above lemmas we obtain Equation (9) by definition of REL:

$$\mathbb{E}_{x_t} \left[ \sup_{c_t} \mathbb{E}_{\hat{y}_t, S_t} \left[ c_t(\hat{y}_t) + \text{REL}(I_{1:t}) \right] \right] \leq \mathbb{E}_{x_t, \rho_t} \left[ \sup_{p_t \in \Delta'_D} \mathbb{E}_{\hat{c}_t, \delta} \left[ \sup_{\pi \in \Pi} \left( 2\delta \hat{c}_t(\pi(x_t)) + A_{\pi,t} \right) \right] \right] + C_t$$

$$\leq \mathbb{E}_{x_t, \varepsilon_t, Z_t, \rho_t} \left[ \sup_{\pi \in \Pi} \left( 2Z_t \cdot \varepsilon_t(\pi(x_t)) + A_{\pi,t} \right) \right] + C_t$$

$$\leq \mathbb{E}_{\rho_{t-1}} \left[ \sup_{\pi \in \Pi} \left( 2Z_t \cdot \varepsilon_t(\pi(x_t)) + A_{\pi,t} \right) \right] + C_t$$

$$\leq \text{REL}(I_{1:t-1}).$$

*Proof of Lemma 9.* For any distribution $p_t \in \Delta'_D$, the distribution of the each coordinate of $\hat{c}_t$ has support on $\{0, \gamma^{-1}K\}$ and is equal to $\gamma^{-1}K$ with probability at most $\gamma/K$. Using $p_t(i)$ to denote $\mathbb{P}\left[\hat{c}_t(i) = \gamma^{-1}K \cdot \mathbf{e}_i\right]$ we can rewrite the LHS (left hand side) of Equation (10) as

$$\sup_{p_t \in \Delta'_D} \mathbb{E}_{\hat{c}_t \sim p_t, \delta} \left[ \sup_{\pi \in \Pi} \left( 2\delta \hat{c}_t(\pi(x_t)) + A_{\pi,t} \right) \right]$$

$$= \sup_{p_t \in \Delta'_D} \left( \left(1 - \sum_i p_t(i)\right) \sup_\pi A_{\pi,t} + \sum_i p_t(i) \mathbb{E}_\delta \left[ \sup_\pi A_{\pi,t} + \frac{2K\delta}{\gamma} \mathbb{1}\left[\pi(x_t) = i\right] \right] \right)$$

$$= \sup_{0 \leq p_t(i) \leq \gamma/K} \left( \left(1 - \sum_i p_t(i)\right) \sup_\pi A_{\pi,t} + \sum_i p_t(i) \mathbb{E}_\delta \left[ \sup_\pi A_{\pi,t} + \frac{2K\delta}{\gamma} \mathbb{1}\left[\pi(x_t) = i\right] \right] \right),$$

where the first equality follows from expanding the expectation with respect to $\hat{c}_t$, and the second equality follows from the definition of $\Delta'_D$. We argue that this value is maximized when each $p_t(i)$ takes on its maximum value, i.e., $\gamma/K$. It suffices to observe that

$$\mathbb{E}_\delta \left[ \sup_\pi A_{\pi,t} + \frac{2K\delta}{\gamma} \mathbb{1}\left[\pi(x_t) = i\right] \right] \geq \sup_\pi \left( A_{\pi,t} + \mathbb{E}_\delta \left[ \frac{2K\delta}{\gamma} \mathbb{1}\left[\pi(x_t) = i\right] \right] \right) = \sup_\pi A_{\pi,t},$$

where the inequality follows from the fact that supremum of expectation is less than expectation of supremum, and the equality uses the fact that $\mathbb{E}[\delta] = 0$. Therefore, we maximize the LHS of Equation (10) via selecting $p_t$ that satisfies $p_t(i) = \frac{\gamma}{K}$ for $i \geq 1$. It follows that

$$\sup_{p_t \in \Delta'_D} \mathbb{E}_{\hat{c}_t \sim p_t, \delta} \left[ \sup_{\pi \in \Pi} \left( 2\delta \hat{c}_t(\pi(x_t)) + A_{\pi,t} \right) \right]$$

$$\leq \left( (1 - \gamma) \sup_\pi A_{\pi,t} + \sum_i \frac{\gamma}{K} \mathbb{E}_\delta \left[ \sup_\pi A_{\pi,t} + \frac{2K\delta}{\gamma} \mathbb{1}\left[\pi(x_t) = i\right] \right] \right)$$

$$= \mathbb{E}_{\varepsilon_t, Z_t} \left[ \sup_{\pi \in \Pi} \left( 2Z_t \cdot \varepsilon_t(\pi(x_t)) + A_{\pi,t} \right) \right],$$

finishing the proof. $\qquad \square$

## 4.2 Proof of Theorem 6

We start with the following standard inequalities for handling the supremum using the moment generating function.

$$\mathbb{E}_{(Z,\varepsilon)_{1:T}}\left[\sup_{\pi\in\Pi}\sum_{i=1}^{T}Z_t\varepsilon_t(\pi(x_t))\right] = \mathbb{E}_{Z_{1:T}}\left[\frac{1}{\lambda}\cdot\mathbb{E}_{\varepsilon_{1:T}}\left[\log\left(\sup_{\pi\in\Pi}e^{\lambda\sum_{t=1}^{T}Z_t\varepsilon_t(\pi(x_t))}\right)\right]\right]$$

$$\leq \mathbb{E}_{Z_{1:T}}\left[\frac{1}{\lambda}\cdot\mathbb{E}_{\varepsilon_{1:T}}\left[\log\left(\sum_{\pi\in\Pi}e^{\lambda\sum_{t=1}^{T}Z_t\varepsilon_t(\pi(x_t))}\right)\right]\right]$$

$$\overset{(i)}{\leq} \mathbb{E}_{Z_{1:T}}\left[\frac{1}{\lambda}\cdot\log\left(\mathbb{E}_{\varepsilon_{1:T}}\left[\sum_{\pi\in\Pi}e^{\lambda\sum_{t=1}^{T}Z_t\varepsilon_t(\pi(x_t))}\right]\right)\right]$$

$$\overset{(ii)}{=} \mathbb{E}_{Z_{1:T}}\left[\frac{1}{\lambda}\cdot\log\left(\sum_{\pi\in\Pi}\prod_{t=1}^{T}\mathbb{E}_{\varepsilon_t}\left[e^{\lambda Z_t\varepsilon_t(\pi(x_t))}\right]\right)\right].$$

Inequality $(i)$ holds due to the concavity of log and $(ii)$ follows from the independence of $\varepsilon_t$. We will additionally assume that $\lambda$ is upper bounded by $\gamma\sqrt{2}/K$ in the remaining analysis.

By our construction of the random variable $\varepsilon_t$, for any fixed $\pi$, $\varepsilon_t(\pi(x_t))$ takes the value $0$ with probability $1 - \frac{1}{K}$ and the values $-1$ and $1$ each with probability $\frac{1}{2K}$. We therefore have that

$$\mathbb{E}_{\varepsilon_t}\left[e^{\lambda Z_t\varepsilon_t(\pi(x_t))}\right] = \left(1 - \frac{1}{K}\right) + \frac{1}{2K}\cdot e^{\lambda Z_t} + \frac{1}{2K}\cdot e^{-\lambda Z_t}$$

$$\leq 1 - \frac{1}{K} + \frac{1}{K}\cdot e^{\lambda^2 Z_t^2/2}$$

$$\leq e^{\frac{1}{K}(e^{\lambda^2 Z_t^2/2}-1)}.$$

The first inequality above uses $e^x + e^{-x} \leq 2e^{x^2/2}$ while the second inequality uses $e^x \geq 1 + x$ for $x \in \mathbb{R}$. This further yields

$$\mathbb{E}_{Z_{1:T}}\left[\frac{1}{\lambda}\cdot\log\left(\sum_{\pi\in\Pi}\prod_{t=1}^{T}\mathbb{E}_{\varepsilon_t}\left[e^{\lambda Z_t\varepsilon_t(\pi(x_t))}\right]\right)\right] \leq \mathbb{E}_{Z_{1:T}}\left[\frac{1}{\lambda}\cdot\log\left(|\Pi|\cdot\prod_{t=1}^{T}e^{\frac{e^{\lambda^2 Z_t^2/2}-1}{K}}\right)\right]$$

$$= \mathbb{E}_{Z_{1:T}}\left[\frac{1}{\lambda}\log(|\Pi|) + \sum_{t=1}^{T}\frac{e^{\lambda^2 Z_t^2/2}-1}{\lambda K}\right]$$

$$= \frac{\log(|\Pi|)}{\lambda} + \frac{1}{\lambda K}\sum_{t=1}^{T}\mathbb{E}_{Z_{1:T}}\left[e^{\lambda^2 Z_t^2/2}-1\right].$$

Recall that $Z_t$ takes the values $0$ and $\frac{K}{\gamma}$ with probabilities $1 - \gamma$ and $\gamma$ respectively. It follows that

$$\mathbb{E}_{Z_{1:T}}\left[e^{\lambda^2 Z_t^2/2}-1\right] = \gamma\left(e^{\frac{\lambda^2 K^2}{2\gamma^2}}-1\right) \leq \gamma\frac{\lambda^2 K^2}{\gamma^2},$$

where the inequality follows from the fact that $e^x - 1 \leq 2x$ for $x \in (0,1)$ and the assumption $\lambda \leq \gamma\sqrt{2}/K$. Therefore,

$$\mathbb{E}_{Z_{1:T}}\left[\frac{1}{\lambda}\cdot\log\left(\sum_{\pi\in\Pi}\prod_{t=1}^{T}\mathbb{E}_{\varepsilon_t}\left[e^{\lambda Z_t\varepsilon_t(\pi(x_t))}\right]\right)\right] \leq \frac{\log(|\Pi|)}{\lambda} + \frac{TK\lambda}{\gamma}.$$

By taking derivative with respect to $\lambda$ we obtain

$$\frac{-\log(|\Pi|)}{\lambda^2} + TK/\gamma = 0,$$

and compute that the equation above is minimized at $\lambda = \sqrt{\frac{\gamma \log(|\Pi|)}{TK}}$. We note that $\lambda$ satisfies the assumption $\lambda \le \gamma\sqrt{2}/K$ because this is equivalent to $\frac{\log(|\Pi|)}{2T} < \frac{\gamma}{K}$, which holds by the assumption of the lemma. Plugging this again yields

$$\log(|\Pi|)\sqrt{\frac{TK}{\gamma \log(|\Pi|)}} + \frac{TK}{\gamma}\sqrt{\frac{\gamma \log(|\Pi|)}{TK}} = 2\sqrt{TK \log(|\Pi|)/\gamma},$$

which is the desired bound.

## 5   Conclusion

In this paper, we presented a novel efficient relaxation for the adversarial contextual bandits problem and proved that its regret is upper bounded by $O(T^{2/3}(K \log |\Pi|)^{1/3})$. This provides a marked improvement with respect to the parameter $K$ as compared to the prior best result and matches the original baseline of Langford and Zhang (2007) for the stochastic version of the problem. As mentioned earlier, non-efficient algorithms can obtain a regret bound of $O(\sqrt{TK \log(|\Pi|)})$, which is information theoretically optimal. While oracle-efficient algorithms can obtain the optimal regret bound in the stochastic setting (Dudik et al., 2011), they do not always achieve optimal regret rates (Hazan and Koren, 2016). Whether or not optimal regret can be obtained in our setting using efficient algorithms remains an open problem and improving both the upper and lower bounds are interesting directions for future work. Additionally, while our work operates in the same setting as prior work (Rakhlin and Sridharan, 2016; Syrgkanis et al., 2016b), it would be interesting to relax some of the assumptions in the setting, most notably the sampling access to the context distribution.

## 6   Acknowledgements

This work is partially supported by DARPA QuICC NSF AF:Small #2218678, and NSF AF:Small #2114269. We thank Alex Slivkins for pointing us to the problem and initial fruitful discussions.

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

## A Proof of Theorem 7

Combining Equation 8 and Theorem 6 we obtain

$$\text{REG}_T \leq 4\sqrt{\frac{TK \log(|\Pi|)}{\gamma}} + \gamma T.$$

Setting the derivative with respect to $\gamma$ of RHS to zero, we obtain

$$T - 2\sqrt{TK \log |\Pi|}\gamma^{-3/2} = 0,$$

which is equivalent to $\gamma = \left(\frac{4K \log |\Pi|}{T}\right)^{1/3}$. Note however that $\gamma$ needs to satisfy $\gamma > KT^{-1}\log(|\Pi|)/2$ and $\gamma \leq 1$. To verify the first condition, we should have $T > K\log(|\Pi|)/(2\gamma)$. Putting our designated $\gamma$ implies that $T$ should satisfy

$$T > K\log(|\Pi|) \cdot (\frac{T}{4K\log(|\Pi|)})^{1/3} \cdot 1/2 \iff T^3 > K^3 \log^3(|\Pi|)\frac{T}{4K\log(|\Pi|)} \cdot 1/8$$

$$\iff T^2 > \frac{K^2 \log^2 |\Pi|}{32} \iff T > \frac{K \log |\Pi|}{4\sqrt{2}},$$

which holds by assumption. To verify the second condition of $\gamma \leq 1$, we need to have $T > 4K\log(|\Pi|)$, which again holds by assumption.

Plugging $\gamma$ to the regret bound, we have

$$\mathcal{O}(T^{2/3}(K\log|\Pi|)^{1/3}),$$

completing the proof.

## B Proof of Lemma 4

The proof is based on Lemma 4 in Syrgkanis et al. (2016b) and is provided for completeness. The pseudocode of our algorithm is provided in Algorithm 2.

---
**Algorithm 2:** Compute $q_t^*$

---
**Input:** value-of-ERM oracle, $(x, \hat{c})_{1:t-1}, x_t$ and $\rho_t$
**Output:** $q_t^*(\rho_t)$ as in Equation 6
Compute for all $i \in [K], \psi_i = \inf_{\pi \in \Pi}$ of

$$\sum_{\tau=1}^{t-1} \hat{c}_\tau(\pi(x_\tau)) + \frac{K}{\gamma}\mathbf{e}_i(\pi(x_t)) + \sum_{\tau=t+1}^{T} 2Z_\tau \varepsilon_\tau(\pi(x_\tau))$$

using the value-of-ERM oracle
Compute $\eta_i = \frac{\gamma(\psi_i - \psi_0)}{K}$ for all $i \in [K]$
Set $m = 1, q = \mathbf{0}$
**for** $k = 1, 2, \ldots, K$ **do**
| $q(i) \leftarrow \min\{(\eta_i)^+, m\}, \quad m \leftarrow m - q(i)$
**end**
If $m > 0$, distribute remaining $m$ uniformly across coordinates of $q$

---

*Proof.* We prove the result by rewriting the minimizer equation to be composed of only calls to our value-of-ERM oracle. For $i \in [K]$, we define $\psi_i$ to be

$$\inf_{\pi \in \Pi} \left( \sum_{\tau=1}^{t-1} \hat{c}_\tau(\pi(x_\tau)) + \gamma^{-1} K\mathbf{e}_i(\pi(x_t)) + \sum_{\tau=t+1}^{T} 2Z_\tau \varepsilon_\tau(\pi(x_\tau)) \right)$$

where $\mathbf{e}_0 = \mathbf{0}$. We can thus write the definition of $q_t^*(\rho_t)$ as

$$\arg \inf_{q \in \Delta_K} \sup_{p_t \in \Delta'_D} \sum_{i=1}^{K} p_t(i) \left( \frac{Kq(i)}{\gamma} - \psi_i \right) - p(0) \cdot \psi_0$$

and note that the values of $\psi_i$ can be computed via a single call to the value-of-ERM oracle, and moreover we require $K + 1$ calls to compute all the $\psi_i$.

Now to compute the minimizer of the above, we first let $z_i = \frac{Kq(i)}{\gamma} - \psi_i$ and $z_0 = -\psi_0$ and rewrite the minimax value as

$$\arg \inf_{q \in \Delta_D} \sup_{p_t \in \Delta'_D} \sum_{i=1}^{K} p_t(i) \cdot z_i + p_t(0) \cdot z_0.$$

We reiterate that each $p_t(i) \leq \gamma/K$ for $i > 0$ and thus we must distribute maximal probability across the $z_i$ coordinates of largest value, i.e. we will put as much probability weight as permitted on $\arg \max_{i \in [K]} z_i$, and proceed to do the same for the second largest value, repeating the process until we exhaust the probability distribution or reach the terminal $z_0$. At this point, we can put the remainder of the probability weight on this final coordinate to ensure summation to 1.

To proceed in analyzing the above water-filling argument, sort the coordinates $z_i$ such that $z_{(1)} \geq z_{(2)} \geq ... \geq z_{(K)}$ for $i > 0$ and further define index $\mu$ to be the smallest index such that $z_{(\mu)} \geq z_0$. By the probability weight distribution argument above, we can reduce the supremum over $p_t$ to be

$$\sum_{i=1}^{\mu} \frac{\gamma}{K} z_{(i)} + (1 - \frac{\gamma}{K}\mu)z_0 = \sum_{i=1}^{\mu} \frac{\gamma}{K}(z_{(i)} - z_0) + z_0.$$

since we maximize $p(i)$ for the $z_{(i)}$ in order and set $p(i) = 0$ for terms less than $z_0$, which would otherwise yield an addition of negative terms in the above summation. Thus, by definition of $\mu$, we have for $i > \mu$

$$\sum_{i=1}^{\mu} \frac{\gamma}{K}(z_{(i)} - z_0) + z_0 = \sum_{i=1}^{K} \frac{\gamma}{K}(z_{(i)} - z_0)^+ + z_0.$$

where the $+$ superscript denotes the RELU operator, $(x)^+ = \max\{x, 0\}$. Therefore, the minimax expression is reformulated as

$$\arg \inf_{q \in \Delta_D} \sum_{i=1}^{K} \frac{\gamma}{K}(z_{(i)} - z_0)^+ + z_0.$$

which is equivalent to minimizing the RELU term

$$\arg \inf_{q \in \Delta_D} \sum_{i=1}^{K} \frac{\gamma}{K}(z_{(i)} - z_0)^+ = \arg \inf_{q \in \Delta_D} \sum_{i=1}^{K} \left( q(i) - \frac{\gamma}{K} \cdot (\psi_i - \psi_0) \right)^+.$$

Simplify notation by setting $\eta_i = \frac{\gamma(\psi_i - \psi_0)}{K}$ so that the expression becomes

$$\arg \inf_{q \in \Delta_D} \sum_{i=1}^{K} (q(i) - \eta_i)^+.$$

We argue the minimization procedure as follows: select $i \in [K]$ such that $\eta_i \leq 0$. Any positive probability weight on $q(i)$ will yield an increase in the expression, whereas for $\eta_i > 0$ we experience no increase in the objective until the value of $\eta_i$ surpasses $q(i)$. Thus, the minimizer will weight the actions with $\min\{\sum_{i:\eta_i>0} \eta_i, 1\}$ on the coordinates with $\eta_i > 0$ and distribute the remaining weight (if nonzero) arbitrarily among $[K]$. This is more precisely outlined in the pseudocode of Algorithm 2. $\qquad \square$

## C Proof of Lemma 8

*Proof of Lemma 8.* Denote by $q_t^* = \mathbb{E}_{\rho_t}[q_t^*(\rho_t)]$. We note that since we are drawing from $\rho_t$, calculating $q_t^*(\rho_t)$, and then drawing from $q_t^*(\rho_t)$, our algorithm is effectively sampling from $q_t^*$, *even though we do not calculate $q_t^*$ explicitly.* Now note that

$$\mathbb{E}_{\hat{y}_t \sim q_t}[c_t(\hat{y}_t)] = \langle q_t, c_t \rangle \leq \langle q_t^*, c_t \rangle + \gamma \left\langle \frac{1}{K}, c_t \right\rangle \leq \mathbb{E}_{\hat{y}_t, \hat{c}_t}[\langle q_t^*, \hat{c}_t \rangle] + \gamma.$$

holds by definition of $q_t(\rho_t)$ and the assumed bounds on $c_t$. Thus, it further holds that

$$\sup_{c_t \in [0,1]^K} \mathbb{E}_{\hat{y}_t, \hat{c}_t}[c_t(\hat{y}_t) + \text{REL}(I_{1:t})] \leq \gamma + \sup_{c_t \in [0,1]^K} \mathbb{E}_{\hat{y}_t, \hat{c}_t}[\langle q_t^*, \hat{c}_t \rangle + \text{REL}(I_{1:t})].$$

By expansion of the second term on the right hand side, we rewrite the relation as

$$\sup_{c_t \in [0,1]^K} \mathbb{E}_{\hat{y}_t, \hat{c}_t}[\langle q_t^*, \hat{c}_t \rangle + \text{REL}(I_{1:t})]$$

$$= \sup_{c_t \in [0,1]^K} \mathbb{E}_{\hat{y}_t, \hat{c}_t}[\langle q_t^*, \hat{c}_t \rangle + \mathbb{E}_{\rho_t}[R((x, \hat{c})_{1:t}, \rho_t)]]$$

$$= \sup_{c_t \in [0,1]^K} \mathbb{E}_{\hat{y}_t, \hat{c}_t}\left[\langle q_t^*, \hat{c}_t \rangle + \mathbb{E}_{\rho_t}\left[\sup_{\pi \in \Pi}(-\hat{c}_t(\pi(x_t)) + A_{\pi,t})\right]\right] + \gamma(T - t)$$

$$= \sup_{c_t \in [0,1]^K} \mathbb{E}_{\hat{y}_t, \hat{c}_t}\left[\mathbb{E}_{\rho_t}\left[\langle q_t^*(\rho_t), \hat{c}_t \rangle + \sup_{\pi \in \Pi}(-\hat{c}_t(\pi(x_t)) + A_{\pi,t})\right]\right] + \gamma(T - t).$$

Furthermore, the symmetric construction of $\hat{c}_t$ implies that $\hat{c}_t$ takes the value $Ke_i/\gamma$ with probability at most $\gamma/K$. Therefore, we know that $\hat{c}_t$ is sampled from a $p_t \in \Delta_D'$ where $\Delta_D'$ is defined as in Equation (5). Taking the maximum over *all possible* $p_t$, we bound the supremum term above with

$$\sup_{c_t \in [0,1]^K} \mathbb{E}_{\hat{y}_t, \hat{c}_t}\left[\mathbb{E}_{\rho_t}\left[\langle q_t^*(\rho_t), \hat{c}_t \rangle + \sup_{\pi \in \Pi}(-\hat{c}_t(\pi(x_t)) + A_{\pi,t})\right]\right]$$

$$\leq \sup_{p_t \in \Delta_D'} \mathbb{E}_{\hat{c}_t \sim p_t}\left[\mathbb{E}_{\rho_t}\left[\langle q_t^*(\rho_t), \hat{c}_t \rangle + \sup_{\pi \in \Pi}(-\hat{c}_t(\pi(x_t)) + A_{\pi,t})\right]\right]$$

$$\leq \mathbb{E}_{\rho_t}\left[\sup_{p_t \in \Delta_D'} \mathbb{E}_{\hat{c}_t \sim p_t}\left[\langle q_t^*(\rho_t), \hat{c}_t \rangle + \sup_{\pi \in \Pi}(-\hat{c}_t(\pi(x_t)) + A_{\pi,t})\right]\right].$$

where the second inequality is an application of Jensen's inequality. We first replace the optimized $q_t^*$ by the corresponding infimum operator over $q_t$, thus the term inside the expectation is equivalent to

$$\inf_{q \in \Delta_K} \sup_{p_t \in \Delta_D'} \mathbb{E}_{\hat{c}_t \sim p_t}\left[\langle q, \hat{c}_t \rangle + \sup_{\pi \in \Pi}(-\hat{c}_t(\pi(x_t)) + A_{\pi,t})\right].$$

conditioned on $\rho_t$. By the minimax theorem, we can interchange the infimum and supremum operators without decreasing the quantity, and this is further upper bounded as

$$\inf_{q \in \Delta_K} \sup_{p_t \in \Delta_D'} \mathbb{E}_{\hat{c}_t \sim p_t}\left[\langle q, \hat{c}_t \rangle + \sup_{\pi \in \Pi}(-\hat{c}_t(\pi(x_t)) + A_{\pi,t})\right]$$

$$= \sup_{p_t \in \Delta_D'} \inf_{q \in \Delta_K} \mathbb{E}_{\hat{c}_t \sim p_t}\left[\langle q, \hat{c}_t \rangle + \sup_{\pi \in \Pi}(-\hat{c}_t(\pi(x_t)) + A_{\pi,t})\right].$$

and moreover, since the objective is linear with respect to $q_t$ we can instead work with

$$\sup_{p_t \in \Delta_D'} \min_{i \in [K]} \mathbb{E}_{\hat{c}_t \sim p_t}\left[\hat{c}_t(i) + \sup_{\pi \in \Pi}(-\hat{c}_t(\pi(x_t)) + A_{\pi,t})\right].$$

Symmmetrization permits us to rewrite the above as

$$\sup_{p_t \in \Delta'_D} \min_{i \in [K]} \mathbb{E}_{\hat{c}_t \sim p_t} \left[ \hat{c}_t(i) + \sup_{\pi \in \Pi} \left( -\hat{c}_t(\pi(x_t)) + A_{\pi,t} \right) \right]$$

$$= \sup_{p_t \in \Delta'_D} \mathbb{E}_{\hat{c}_t \sim p_t} \left[ \sup_{\pi \in \Pi} \left( \min_{i \in [K]} \mathbb{E}_{\hat{c}'_t \sim p_t} [\hat{c}'_t(i)] - \hat{c}_t(\pi(x_t)) + A_{\pi,t} \right) \right]$$

$$\leq \sup_{p_t \in \Delta'_D} \mathbb{E}_{\hat{c}_t \sim p_t} \left[ \sup_{\pi \in \Pi} \left( \mathbb{E}_{\hat{c}'_t \sim p_t} [\hat{c}'_t(\pi(x_t))] - \hat{c}_t(\pi(x_t)) + A_{\pi,t} \right) \right]$$

$$\leq \sup_{p_t \in \Delta'_D} \mathbb{E}_{\hat{c}_t, \hat{c}'_t \sim p_t} \left[ \sup_{\pi \in \Pi} \left( \hat{c}'_t(\pi(x_t)) - \hat{c}_t(\pi(x_t)) + A_{\pi,t} \right) \right].$$

where the last inequality is an additional application of Jensen's inequality. Since $\hat{c}_t, \hat{c}'_t$ are sampled from the same distribution, this can be rewritten as

$$\sup_{p_t \in \Delta'_D} \mathbb{E}_{\hat{c}_t, \hat{c}'_t \sim p_t} \left[ \sup_{\pi \in \Pi} \left( \hat{c}'_t(\pi(x_t)) - \hat{c}_t(\pi(x_t)) + A_{\pi,t} \right) \right]$$

$$= \sup_{p_t \in \Delta'_D} \mathbb{E}_{\hat{c}_t, \hat{c}'_t \sim p_t, \delta} \left[ \sup_{\pi \in \Pi} \left( \delta(\hat{c}'_t(\pi(x_t)) - \hat{c}_t(\pi(x_t))) + A_{\pi,t} \right) \right]$$

$$= \sup_{p_t \in \Delta'_D} \mathbb{E}_{\hat{c}_t, \hat{c}'_t \sim p_t, \delta} \left[ \sup_{\pi \in \Pi} \left( \delta \hat{c}'_t(\pi(x_t)) - \delta \hat{c}_t(\pi(x_t)) + A_{\pi,t} \right) \right].$$

and further split the term within the inner supremum to obtain

$$\sup_{p_t \in \Delta'_D} \mathbb{E}_{\hat{c}_t, \hat{c}'_t \sim p_t, \delta} \left[ \sup_{\pi \in \Pi} \left( \delta \hat{c}'_t(\pi(x_t)) - \delta \hat{c}_t(\pi(x_t)) + A_{\pi,t} \right) \right]$$

$$= \sup_{p_t \in \Delta'_D} \mathbb{E}_{\hat{c}_t, \hat{c}'_t \sim p_t, \delta} \left[ \sup_{\pi \in \Pi} \left( \delta \hat{c}'_t(\pi(x_t)) + \frac{A_{\pi,t}}{2} - \delta \hat{c}_t(\pi(x_t)) + \frac{A_{\pi,t}}{2} \right) \right]$$

$$\leq \sup_{p_t \in \Delta'_D} \mathbb{E}_{\hat{c}_t, \hat{c}'_t \sim p_t, \delta} \left[ \sup_{\pi \in \Pi} \left( \delta \hat{c}'_t(\pi(x_t)) + \frac{A_{\pi,t}}{2} \right) + \sup_{\pi \in \Pi} \left( -\delta \hat{c}_t(\pi(x_t)) + \frac{A_{\pi,t}}{2} \right) \right]$$

$$= \sup_{p_t \in \Delta'_D} \mathbb{E}_{\hat{c}_t \sim p_t, \delta} \left[ \sup_{\pi \in \Pi} \left( 2\delta \hat{c}_t(\pi(x_t)) + A_{\pi,t} \right) \right].$$

$\square$

# D  Proof of the final step condition

**Lemma 10.** *The relaxation function* (4) *satisfies the final step condition* (2).

*Proof.* By setting $t = T$ in our relaxation $\text{REL}(I_{1:T})$ and definition of our unbiased estimator $\hat{c}_t$ of $c_t$,

$$\mathbb{E}_{\hat{y}_{1:T}} \left[ \text{REL}(I_{1:T}) \right] = \mathbb{E}_{\hat{y}_{1:T}} \left[ -\sup_{\pi \in \Pi} \sum_{\tau=1}^{T} \hat{c}_\tau(\pi(x_\tau)) \right]$$

$$\geq \sup_{\pi \in \Pi} -\mathbb{E}_{\hat{y}_{1:T}} \left[ \sum_{\tau=1}^{T} \hat{c}_\tau(\pi(x_\tau)) \right]$$

$$= \sup_{\pi \in \Pi} -\sum_{\tau=1}^{T} c_\tau(\pi(x_\tau)).$$

$\square$

