# OpenReview forum: "An Improved Relaxation for Oracle-Efficient Adversarial Contextual Bandits"
_NeurIPS.cc/2023/Conference — NeurIPS 2023 poster_

### Official Review · Reviewer_2pq4 · 2023-06-14

**Soundness:** 3 good
**Presentation:** 2 fair
**Contribution:** 3 good
**Rating:** 7
**Confidence:** 4

**Summary:**

This paper investigates the contextual bandit problem where the contexts are sequentially sampled from a known i.i.d. source, and losses are generated adversarially. The primary focus is on developing oracle-efficient algorithms that minimize the expected regret (the expectation over both the context's randomness and the algorithm's internal randomness). Specifically, the paper demonstrates that for any finite policy set $\Pi$ with an value-ERM oracle (capable of finding the minimal policy loss given any sequence of losses), one can achieve an oracle-efficient regret bound of the order $T^{2/3}(K\log|\Pi|)^{1/3}$, where $K$ is the number of arms. This finding surpasses the previous bound of  order $(TK)^{2/3}(\log |\Pi|)^{1/3}$ in Syrgkanis (2016).

The proof technique essentially follows the relaxation-based argument from Rakhlin and Sridharan (2016) and Syrgkanis (2016). However, a distinguishing feature of this paper is the introduction of novel Rademacher vectors when defining the relaxation. The authors successfully incorporate these new Rademacher vectors into the analysis framework established by Syrgkanis (2016), thereby advancing the current state of the art.

**Strengths:**

The main strength of this paper is the oracle-efficient regret bound that improves the state of the art. The authors also introduces several new ideas and techniques that is of independent interests.

**Weaknesses:**

The submission does not seem to have any significant weaknesses. However, I have a few minor comments concerning the presentation:

1. The paper contains several typographical errors, including:
    - line 1: "for for"
    - line 284: isn't  (ii) follows from the independence of $\epsilon_t$?
    - line 394: should be "Proof of Lemma 4"

2. It would be better if the authors could reference the corresponding appendix section in the main text where proof is provided.

3. For a more direct comparison with previous bounds, it would be beneficial to consider some specific values for $K$, such as setting $K=T^{\alpha}$, where $\alpha<1$. This would provide more tangible examples and help in understanding the practical implications of the bounds.

**Questions:**

I have a couple of questions for the authors:

1. The paper currently focuses on the finite policy set. Could the techniques developed in this paper be utilized to enhance the Rademacher complexity-based bounds, as discussed in Rakhlin and Sridharan (2016)?

2. Could the i.i.d. assumption be relaxed? For instance, could it accommodate smooth adversaries as proposed in Haghtalab et al. (2022)?

**Limitations:**

No issue with negative societal impact.

---

> ### Author Rebuttal · Authors · 2023-08-09
>
> Thank you for your comments.
> We are glad that
> you think our ideas and techniques are of independent interests.
> Please find below our response to the questions and comments mentioned in your review.
>
> **Questions**
>
> Q1: We initially obtain a "Rademacher style" bound (Equation 8 in our paper) in our proofs,
>   but we upper bound this using Theorem 6.
>   The bounds in Equation 8 are similar to the bounds of Rakhlin and Sridharan (Equation 8 in their paper) but
>   do not seem to be directly comparable given the existence of the $Z_t$ term (see Equation 4 of their paper for their definition of $\mathcal{R}$).
>
> Q2: Our main focus in this paper was improving the existing regret bounds in
>   the setting based on prior work as the problem is already difficult.
>   We agree however that the mentioned relaxation of the assumption is an interesting direction
>   for future work and will add a discussion in the conclusion section for our revision.
>   (see also "Relaxing the assumption of the setting" in our general response)
>
> **Weaknesses**
>
> > The paper contains several...
>
> Thank you for pointing out these errors. We will fix them for the revised version of our paper.
>
> > It would be better if the authors...
>
> We agree that this helps improve the readability of the paper and will add the section numbers for the revised version of our paper.
>
> > For a more direct comparison with previous bounds,...
>
> Thanks for a great suggestion. By setting $K = T^{\alpha}$, the suggested
> bounds by Rahklin and Sridharan 2016 and independently by Syrgkanis et al. 2016a
> equal $O(T^{3/4+\alpha/2} \log(|\Pi|)^{1/4})$. This is further improved by
> Syrgkanis et al. 2016b to the ratio of $O(T^{2/3+2/3\cdot \alpha}
> \log(|\Pi|)^{1/3})$, which has been state-of-the-art before our work. Our
> technique guarantees $O(T^{2/3+1/3\cdot \alpha} \log(|\Pi|)^{1/3})$. For
> example, putting $\alpha = 1/2$ yields a vague regret bound of $O(T
> \log(|\Pi|)^{1/3})$ for Syrgkanis et al. 2016b, whereas we obtain $O(T^{5/6}
> \log(|\Pi|)^{1/3})$ which is still sub-linear for $K = T^{1/2}$ with respect to
> $T$. Moreover, the upper bound of Syrgkanis et al. 2016b hold upon the
> condition $T \ge K^2 \log (|\Pi|)$, thus their result does not apply if $K =
> \Omega(T^{1/2})$.

---

> > ### Comment · Reviewer_2pq4 · 2023-08-10
> > **no additional questions**
> >
> > Thank you for addressing my questions. I have no additional questions at this point.

---

### Official Review · Reviewer_gZc2 · 2023-06-15

**Soundness:** 3 good
**Presentation:** 4 excellent
**Contribution:** 3 good
**Rating:** 6
**Confidence:** 3

**Summary:**

In this paper, the authors consider a classic contextual bandit problem with adversarial loss and stochastic context. Specifically, they proposed a relaxation-based algorithm which achieves $O(K^{1/3}T^{2/3}\log^{1/3}|Pi|)$ expected regret bound, improving upon the best known $O(K^{2/3}T^{2/3}\log^{1/3}|Pi|)$ obtained by Syrgkanis et al., 2016 under the same assumption. The algorithm is oracle-efficient, which requires $K+1$-number of call to the ERM oracle. The main improvement upon the algorithm proposed in [Syrgkanis et al., 2016] is a new relaxation expression. Specifically, they replace the random vector $\epsilon_t$ with all entries a Radamacher random variable by a random vector with a uniformly chosen entry being the Radamacher random variable. The reason for this improvement I think is due to the fact that the loss for each action has a uniform $\frac{\gamma}{K}$ probability to be observed and recorded in the loss estimator construction, which is missed in the construction of [Syrgkanis et al., 2016].

**Strengths:**

- This paper considers a classic contextual bandit problem with a provable better theoretical guarantee in the expected regret bound.
- The writing of this paper is clear.
- The proposed algorithm is clear, which is mainly based on the relaxation-based algorithm proposed in [Syrgkanis et al., 2016] but with a better construction on the relaxation function $Rel$.
- The proofs also look correct to me. Although the modification compared with [Syrgkanis et al., 2016] is not that much, the new construction on this relaxation function looks interesting to me.

**Weaknesses:**

- One concern is the significance of the obtained results. Compared with [Syrgkanis et al., 2016], the improvement is a factor of $K^{1/3}$. Although the author argued that this improvement can be significant when considering continuous action space with a discretization, I think the order on $T$ may be more important. The main difficulty from obtaining the $\sqrt{T}$ regret bound seems to be the fact that the loss estimator is meaningful only when the algorithm explore, which also appears in the BISTRO+ algorithm proposed in [Syrgkanis et al., 2016].
- Minor part: I think the algorithm actually does not require the context distribution to be known but instead require sampling from the context distribution. So I think the abstract description is not accurate, meaning that the result is stronger.
- Minor typo:
(1) Line 160: minimizes -> minimize
(2) Line 266-267: missing a right bracket



**Questions:**

- I wonder whether it is possible to further generalize this relaxation-based algorithm to achieve $O(\sqrt{T})$ type result?
- In addition, the current algorithm requires fresh samples of the stochastic context distribution, which is the same as what is assumed in [Syrgkanis et al., 2016]. I wonder whether it is possible to remove this assumption, since in some applications, this assumption may not hold.


**Limitations:**

See Weakness and Questions.

---

> ### Author Rebuttal · Authors · 2023-08-09
>
> Thank you for your comments. We are glad that you found the writing of our paper to be clear.
> Please find below our response to the questions and comments mentioned in your review.
>
> **Questions**
>
> Q1: We note that
>   the regret lower bound of $\sqrt{TK\log(|\Pi|)}$ holds for all (not necessarily efficient) algorithms.
>   It is not known whether such a regret bound can be obtained efficiently
>   as oracle-efficient algorithms
>   do not always obtain optimal regret rates (Hazan and Koren, 2016).
>   Whether or not this is the case for our problem remains an open problem.
>   We will further expand on our discussions of this point in the conclusion section.
>   (see also "Gap between upper and lower bounds" in our general response)
>
> Q2:
> Our focus on this paper was improving the existing regret bounds in
>   the setting based on prior work as the problem is already difficult.
>   We agree however that the mentioned relaxation of the assumption is an interesting direction
>   for future work and will add a discussion in the conclusion section for our revision.
>   (see also "Relaxing the assumption of the setting" in our general response)
>
> **Weaknesses**
>
> > One concern is the significance of the obtained results,
>
> Please see "Importance of the improvements" in our general response
>
> > I think the algorithm actually does not require the context distribution to be known but instead require sampling from the context distribution
>
> Thank you for pointing this out. This is correct and we will further emphasize this in the paper.
>
> Typos:
> Thank you for pointing out the typos; we will fix them for our revision.

---

> > ### Comment · Reviewer_gZc2 · 2023-08-11
> > **Thanks**
> >
> > Thank you for your response to my questions. Through I still think the improvement over K is not that significant (from order 2/3 to order 1/3), I agree that the technique used in the analysis is interesting and I keep my original score.

---

### Official Review · Reviewer_yx68 · 2023-06-17

**Soundness:** 4 excellent
**Presentation:** 4 excellent
**Contribution:** 2 fair
**Rating:** 6
**Confidence:** 5

**Summary:**

The paper studies the problem of minimizing regret for online adversarial contextual bandits using an ERM oracle. If one doesn't care about oracle efficiency, then the well known EXP4 algorithm achieves the optimal rate. However, the best previous oracle-efficient result is due to [Syrgkanis et al. '16]. This paper improves a factor of $K^{1/3}$ in the regret for this problem over prior work, where $K$ is the number of arms. The algorithm uses $O(K)$ calls to the oracle in every round.

Their proof relies on a new relaxation under the relax-and-randomize framework of [Rakhlin-Sridharan] which has reduced variance compared to prior works.

**Strengths:**

- The paper is exceptionally well written. The problem statement, relationship to prior works, explanation of the main contributions, and technical aspects of the proof are all very clear. As such, it was easy for me to follow and understand.
- The relaxation function that is used to improve the dependence on $K$ seems to be novel.


**Weaknesses:**

- The only weakness is that I'm not sure how significant the contribution/impact of the paper is: it only amounts to a $K^{1/3}$ improvement in the regret. There is still a substantial gap between the upper bound and lower bounds. The paper also doesn't really touch upon whether their rate is improvable for oracle efficient algorithms (we know EXP4 attains the optimal regret if we don't care about oracle efficiency), potentially hinting at some sort of separation result.

**Questions:**

I have checked almost all of the proofs of the paper. I had a few questions, ranging from questions on the technical details to more higher-level questions.

1. What is the oracle complexity for prior works? Can some discussion on this be included (if it is interesting)?
2. What can be done if the learner does not have sample access to the distribution $\mathcal{D}$? Is learning possible here? EXP4 does not require this additional assumption.
3. Line 191: what is meant by the "discretization scheme"?
4. Any intuition for what the parameter $\gamma$ controls? Should we expect any improvements in the final bound with a changing $\gamma_t$?
5. While I understand the construction of the relaxation function (specifically the design of $\epsilon_t$ and $Z_t$), it is still quite mysterious. I see how it is used in Lemma 9. Perhaps the authors could include more intuition on this.
6. In theorem 6, what is the quantifier over the $x_t$? Are we also taking expectation wrt $x_t$?
7. writing comment: In Lemma 9, perhaps $\delta$ should be properly defined. It is defined in the previous lemma, but as is, Lemma 9 cannot be read "independently" Maybe state the definition of $\delta$ before the statements of Lemma 8 and 9?
8. writing comment: In appendix, you write Theorem 4, but in the main text, it is stated as Lemma 4.
9. In line 398, the derivation here could include more explanation.
10. line 403: "We reiterate that each $p_t(i) \le \gamma$" I thought by construction we had $p_t(i) \le \gamma/K$, so we can't put $\gamma$ mass on any arm $i\in [K]$ (as stated in line 211). Am I missing something here?
11. line 432: I did not understand why the probability of $\hat{c}_t$ taking the value $Ke_i/\gamma$ is at most $\gamma/K$. By Eq (6), don't we have the probability is at least $\gamma/K$?

More speculative, for my own understanding:
- Do you think that this relax/randomize strategy has limitations? It seems like we cannot substantially improve this bound to get $\sqrt{T}$-style regret.
- Can the arguments from [Hazan-Koren '16] be used to get lower bounds against oracle-efficient algorithms?
- Superficially, the guarantee that you get seems to be what one would expect if they adapted a "PAC" algorithm that got $K/\epsilon^2$ sample complexity using an online2batch conversion (of course, here in the adversarial setting "PAC" doesn't make sense because the costs are adversarial). Does this hint at a limitation of oracle-efficient algorithms?

**Limitations:**

None.

---

> ### Author Rebuttal · Authors · 2023-08-09
>
> Thank you for your comments.
> We are glad that you found our paper to be "exceptionally well written".
> Please find below our response to the questions and comments mentioned in your review.
>
> **Questions**
>
> Q1: The oracle complexity is equivalent to that of Syrgkanis et al. 2016 and
> our main result is in the improved relaxation, not the leveraged
> oracle. We will make this more explicit in the final version of our paper.
>
> Q2: Currently our techniques rely on knowing D for constructing the
> hallucinated reward vectors and it is unclear if this can be easily relaxed. We
> believe that obtaining results that relax the assumption, even with a
> potentially worse regret rate, is an interesting direction for future research and will
> discuss this in the conclusion section (see also "Relaxing the assumption of the setting" in our general response).
> We note that the sampling access assumption is also made in both of the papers
> that operate in our setting (Rakhlin and Sridharan, 2016; Syrgkanis et al.
> 2016).
>
> Q3: Discretization here refers to the approximation of a continuous set
> with a discrete set (e.g., see chapter 8.2 in Slivkins (2019)).
>
> Q4: The main purpose of $\gamma$ is to ensure that each arms is pulled with some
> non-negligible probability and, through Equation (3), the parameter intuitively
> affects the variance of the rewards. Our current techniques do not seem to
> benefit from varying $\gamma$ with time as we require the same bound on the variance of
> rewards through time.
>
> Q5:
> We will expand on the intuition of these variables
> for both our novel aspects and the aspects based on prior work
> in our revision.
> Due to the space constraints of the rebuttal, we focus on the novelty aspect here.
> The intuition of the change has two parts: the admissibility of the change
> and why it helps obtain a better regret bound:
> - Admissibility: The main intuition for why only a single Rademacher variable is sufficient
>   for ensuring admissibility
>   is that the symmetrization step of the Relax and randomize framework
>   applies only to a single (random) action (see Lines 440-443 as well as the LHS
>   of Lemma 9). Applying noise to all the entries leads to valid upper bound (as is done
>   by prior work), but is not tight.
>   The main reason these works apply this bound is (in our understanding) that the distribution
>   of the noise is not known (as it depends on $c$ and the algorithm), and therefore they
>   take the upper bound of applying it everywhere.
>   Our main insight here is that knowing this distribution is not necessary.
>   Since the variance of the reward vector is maximized when
>   $\hat{c}_t$ is largest, we can effectively consider the case of $c_t$ taking its
>   maximum value (as formalized in lines 273-277).
>   Focusing on this case however, the distribution of $\hat{c}_t$ is known because,
>   regardless of what $q_t$ is, it takes the value
>   $K\gamma^{-1}e_i$ with probability $\gamma / K$ (see Equation 3).
> - Why it helps: The reason the change helps can be seen by looking at the LHS of Theorem 6.
>   Consider fixed values for $Z_{1:T}$.
>   If we put random noise on all the coordinates, then the supremum
>   leads to a policy function $\pi$ that assigns $x_t$ to coordinates
>   with noise values equal to 1. By restricting the noise, we are restricting
>   the power of the supremum.
>
> Q6: The values of $x_t$ are fixed; since the claim holds for any fixed $x_t$, it also
>   holds for a random choice of $x_t$ by iterated expectation.
>
> Q7, 8: Thank you for pointing out these issues; we will apply your comments in the revised version of our paper.
>
> Q9: The rewrite is more easily seen by recalling the definition of
> $q_t^*(\rho_t)$ from line 213 and expanding the expectation while substituting
> the equation on line 201 for $R((x, c_t)_{1:t}, \rho_t)$ and further leveraging
> the definition of $\psi_i$ from line 397. We will expand on this substitution
> of variables for the full version of our result to be more clear.
>
> Q10: Thank you for pointing out this typo. The upper bound on $p_t(i)$ should be $\gamma / K$ as you correctly noted.
>   We will fix this in our revision.
>
> Q11: Equation 6 ensures that the action $i$ is chosen with probability at least $\gamma/K$. However,
>   choosing the $i$-th action does not necessarily lead to $\hat{c}_i$ being set to
>   $Ke_i/\gamma$ however because of the extra noise in Equation (3).
>   Given this Equation, the probability of $\hat{c}_t(i)=Ke_i/\gamma$ for
>   $\hat{y}_t \sim q_t$ equals $c_t(i) \gamma/K$ which is at most $\gamma/K$ given
>   $c_t(i) \le 1$.
>
> Lower bounds / Limitations of oracle-efficient algorithms:
> Our intuition is similar and
>   it seems to us that at least the existing techniques used for relax and randomized are insufficient
>   for improving the bound.
>   While we were not able to prove any improved lower bounds, we consider this to be an important direction for future work and
>   we agree that the arguments from [Hazan and Koren] are a
>   good starting point for exploring this direction.
>   (see also "Gap between upper and lower bounds" in our general response)
>
> **Weaknesses**
>
> > not sure how significant the contribution/impact ... only amounts to a $K^{1/3}$ improvement in the regret
>
> Please see "Importance of the improvements" in our general response.
>
> > There is still a substantial gap between the upper bound and lower bounds … potentially hinting at some sort of separation result.
>
> Please see "Gap between upper and lower bounds" in our general response.

---

> > ### Comment · Reviewer_yx68 · 2023-08-14
> > **Thanks**
> >
> > Thank you for your response. I have no further questions.

---

### Official Review · Reviewer_D6vD · 2023-07-04

**Soundness:** 4 excellent
**Presentation:** 4 excellent
**Contribution:** 4 excellent
**Rating:** 7
**Confidence:** 3

**Summary:**

This paper proposed a new algorithm for Oracle-Efficient Adversarial Contextual Bandits that achieves the best known regret bound and improve upon the previous best bound in its dependency on the action set size $K$.

**Strengths:**

The algorithm is elegant. The paper is well-written and pleasant to read. The theoretical result is sound.

**Weaknesses:**

The theoretical results are sound, but I believe for any work that make effort towards some type of computational efficiency, oracle-efficiency being one of them, experimental evaluation is always appreciated.

**Questions:**

NA

---

> ### Author Rebuttal · Authors · 2023-08-09
>
> Thank you for your comments. We are glad that you found the paper well-written and that you believe our algorithm is elegant.
> Please find below our response to the questions and comments mentioned in your review.
>
> > any work that make effort towards some type of computational efficiency ... experimental evaluation is always appreciated.
>
> Our main focus on this paper was
> on the theoretical aspects of the problem.
> We note that the other existing works for adversarial contextual bandits
> (Rakhlin and Sridharan, 2016; Syrgkanis et al. 2016a, Syrgkanis et al., 2016b),
> also do not present experimental evaluation. Given the similarities between our
> frameworks, we intuitively expect our algorithm to outperform Bistro
> (Rakhlin and Sridharan 2016) and Bistro+ (Syrgkanis et al. 2016b) in practice
> since our main difference is in the construction of the relaxation function and
> we reduce the variance of the Rademacher vectors in each round.
> We agree however that exploring the practical challenges of the problem and comparing
> algorithms designed for different settings (stochastic, adversarial, bounded difference, non-stationary, ... )
> is an important direction for future work and will further emphasize this point in the conclusion section.

---

### Author Rebuttal · Authors · 2023-08-09

We thank the reviewers for their comments.
We here respond to the concerns that were shared by
more than one reviewer.

**Relaxing the assumption of the setting (Reviewers yx68, gZc2, and 2pq4)**

Our focus on this paper was improving the existing regret bounds in
the setting based on prior work (Rakhlin and Sridharan, 2016; Syrgkanis et al., 2016) because the problem is already difficult.
As pointed out the reviewers, the problem setting
can be generalized
by removing the sample access to the context distribution.
We agree that this is an interesting direction for future work
and we will discuss this in the conclusion section for our revision.

**Importance of the improvements (Reviewers yx68 and gZc2)**

In our view, the improved dependence on the number of arms (i.e., $K$)
is significant because $K$ can be large in many applications
such as recommender systems.
Additionally, the value can be large if the bandit algorithm
is used as part of a *reduction* that defines "fake" or "synthetic" arms for a given
problem.
In our view, for large values of $K$, our improvement can be very significant.
Indeed, if we consider the regime of $K=T^{\alpha}$ as suggested by Reviewer
2pq4, then we can see that our results can still imply sub-linear regret bounds for
$\alpha > 1/2$, while the results in prior work cannot.
Finally, given the importance of obtaining oracle-efficient algorithms as
evidenced by the large body of work (see lines 44-50)
and the lack of progress for the problem after
the work of Syrgkanis et al. (2016), we consider our result to be significant.

**Gap between upper and lower bounds (Reviewers yx68 and gZc2)**

Currently, there is a gap between the upper and lower bounds known for our problem.
The best upper bound is $O(T^{2/3}(K\log(\Pi))^{1/3})$ as shown by our paper while the best known lower bound is
$\Omega(\sqrt{TK\log(\Pi)})$ which holds
for all (including non-efficient) algorithms.
This lower bound may not be tight
for oracle-efficient algorithms however as these algorithms do not always obtain optimal regret rates (Hazan and
Koren, 2016).
As mentioned correctly by the reviewers, there can potentially be be a strict
separation in the regret rates obtained by efficient and non-efficient
(e.g., EXP4) algorithms for adversarial contextual bandits.
Whether or not this
is the case remains open.
We briefly discuss this in the end of Section 2 and the conclusion section and will further expand on our discussion for our revised version.

---

### Decision · Program_Chairs · 2023-09-21

**Decision:**

Accept (poster)

**Comment:**

All reviewers agree that the paper makes a substantial theoretical contribution to an open problem in the area of adversarial contextual bandits.